# A Fabry-Pérot cavity coupled surface plasmon photodiode for electrical biomolecular sensing

Giles Allison [1✉], Amrita Kumar Sana [1], Yuta Ogawa[1], Hidemi Kato[1], Kosei Ueno [2], Hiroaki Misawa [3,4], Koki Hayashi[1] & Hironori Suzuki [1]

Surface plasmon resonance is a well-established technology for real-time highly sensitive label-free detection and measurement of binding kinetics between biological samples. A common drawback, however, of surface plasmon resonance detection is the necessity for far field angular resolved measurement of specular reflection, which increases the size as well as requiring precise calibration of the optical apparatus. Here we present an alternative optoelectronic approach in which the plasmonic sensor is integrated within a photovoltaic cell. Incident light generates an electronic signal that is sensitive to the refractive index of a solution via interaction with the plasmon. The photogenerated current is enhanced due to the coupling of the plasmon mode with Fabry-Pérot modes in the absorbing layer of the photovoltaic cell. The near field electrical detection of surface plasmon resonance we demonstrate will enable a next generation of cheap, compact and high throughput biosensors.

[1] IMRA Japan Co., Ltd., 2-36, Hachiken-cho, Kariya, Aichi 448-8650, Japan. [2] Department of Chemistry, Faculty of Science, Hokkaido University, Kita-ku, Sapporo 060-0810, Japan. [3] Research Institute for Electronic Science, Hokkaido University, Kita-ku, Sapporo 001-0021, Japan. [4] Center for Emergent Functional Matter Science, National Yang Ming Chiao Tung University, Hsinchu 30010, Taiwan. ✉email: giles.allison@imra-japan.com

Changes in lifestyle and personalized healthcare have seen a rapid growth in demand for low cost point of care diagnostic and well-being devices[1]. Surface plasmon resonance (SPR) has proven to be an invaluable technology for biomolecular detection[2] that has been applied to fields as diverse as disease detection[3,4], drug discovery[5], food safety[6,7] and environmental monitoring[8,9]. The recent pandemic caused by the novel coronavirus (SARS-CoV-2) has served as one example to heighten the value of technologies able to rapidly test for the presence of antibodies and antigens[10,11]. Broadly speaking SPR can be split into two categories; propagating surface plasmons (PSPR) in continuous thin metallic films[12] and localized surface plasmons (LSPR) in nanopatterned metallic films[13] and metallic nanoparticles[14]. In both cases, collective oscillations of free electrons on the metal surface are induced by optical fields and these oscillations are sensitive to the refractive index (RI) of the surrounding media. Detection of the plasmon can take the form of optical measurements of reflection or transmission and the detection of the hot-electron currents. Such hot-electrons have further applications that can be exploited for photocatalysis[15,16], photodetection[17], and photon energy conversion[18].

Here we introduce an efficient electrical detection scheme based on PSPR. The optically transparent substrate typical for PSPR is replaced with a semi-transparent amorphous silicon layer that forms an absorbing layer sandwiched between the metallic layer and a transparent conducting electrode. Incident light generates a current driven by the built-in potential of the device in a manner analogous to that of photovoltaic cells that have been shown to have efficiencies up to 12.69%[19,20]. By illuminating a sufficiently thin silicon layer with monochromatic visible light at a resonant angle it is possible to excite a plasmon in the metal layer while simultaneously generating a current through the device. The plasmon acts to disrupt the electric field distribution within the silicon layer resulting in drastically reduced photocurrent. We further enhance and amplify this phenomenon by utilizing the silicon layer synergistically as an optical cavity to generate Fabry-Pérot (FP) modes that become coupled to the plasmon mode. We confirm the mechanism through simulations of the electric field and show experimentally the electric detection of the plasmon mode and use this method to detect changes in refractive index (RI) and protein–protein interactions—in this case, antigen–antibody interaction of the SARS-CoV-2 nucleocapsid protein.

## Results

**Working principle.** The operation of the device as a detector for protein–protein interaction is illustrated in Fig. 1a, b. The device is mounted onto a prism in the Kretschmann geometry[21] (Fig. 1a) that is furthermore mounted onto a rotating stage. The angle of rotation is defined such that the metal film is perpendicular to the incoming laser beam at $\theta_{rot} = 0°$ as indicated in the inset to the figure. (Henceforth, unless stated, all angles refer to the angle of rotation - internal angles within the device can be calculated using the formulae given in Supplementary Note 1.) Light passes through the prism, transparent indium-tin-oxide (ITO) electrode, and semi-transparent amorphous silicon layer (transparency data is included in Supplementary Note 2) and reaches the 30 nm thick gold layer. The device operates as a photodiode with the silicon-gold interface supplying the in-built potential difference and the silicon layer serving the dual purpose of optical nanocavity and absorbing, photogenerating semiconductor. The generated photocurrent depends on the wavelength of light, thickness of silicon layer, and angle of illumination. Amorphous silicon was chosen due to its large refractive index & extinction coefficient, aforementioned large optical photogeneration efficiency,

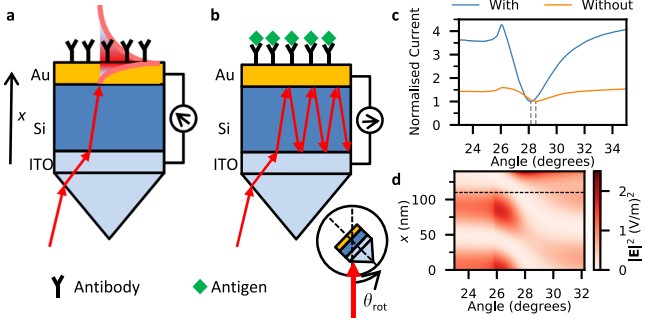

**Fig. 1 Surface plasmon detection and Fabry-Pérot enhancement. a, b** Schematic representation of current generation. Photocurrent is generated by incident light photoexciting electrons in the silicon layer that are subsequently driven into the ITO contact by the in-built electric field. In **a** antibodies are immobilized on the gold surface and the incident angle fixed to generate a surface plasmon resonance in the gold film. Reflection is minimal at this angle, so only incoming light generates photocurrent. In **b** antigens are bound to the antibodies, which causes a change of refractive index leading to a loss of plasmon resonance and increase in reflection by the gold film and subsequently absorption in the silicon layer. Binding is thus observed as a change of current. The silicon layer acts as a nanocavity with multiple reflections enhancing the change of current. The inset indicates the angle of rotation ($\theta_{rot}$) of the device with respect to the laser. **c** Experimental photocurrent as a function of rotation angle with $\lambda = 670$ nm and gold thickness 30 nm for silicon thicknesses 110 nm (blue) and 140 nm (orange). The thinner device supports cavity modes of **b** whereas the thicker device does not. The current is normalized at the SPR angle indicated by the dashed lines. **d** Simulated electric field intensity at $\lambda = 670$ nm as a function of angle and thickness through the device for the second cavity mode (silicon thickness 110 nm) where the dashed line indicates the silicon/gold interface. The plasmon field is observed in the upper portion of the gold layer (125 nm < $x$ < 140 nm) at the SPR angle (~28°) and corresponds to low absorption in the silicon layer. Source data are provided as a Source Data file.

straightforward fabrication and compatibility with state of the art solar cell technologies. The thickness of the silicon layer was chosen such that the optical path length within the layer supports a Fabry-Pérot mode at the angle of incidence for the specified laser wavelength.

We detect the antigen–antibody binding using the following procedure (see methods). First, antibodies (Ab) are immobilized onto the surface of the gold via a linking layer using an amine coupling method[22] and the sample covered with an inert buffer solution (Fig. 1a). The angle of rotation is then fixed at the corresponding SPR angle of this solution, $\theta_{rot} = \theta_{SPR}^{Ab}$. Reflection from the gold film is reduced due to the formation of the plasmon so the majority of the photocurrent is generated along the single optical path of the incoming light. The buffer solution is then replaced with a solution containing an antigen (Ag). Successful antigen–antibody binding (Fig. 1b) significantly changes the RI of the solution in proximity to the surface, which in turn changes the SPR angle resulting in detuning from plasmon resonance ($\theta_{rot} \neq \theta_{SPR}^{Ab+Ag}$). Antigen–antibody binding is thus detected in real time as an increase in photogenerated current due to increased reflection by the gold layer enhanced by multiple internal reflections within the silicon nanocavity.

Alternatively, the device can be operated in angle-sweep mode (Fig. 1c). Here clear angular dependence of the photocurrent is observed similar to the typical reflection measurements of conventional optical SPR biosensors[23]. In this mode of operation changes of RI are detected as shifts of $\theta_{SPR}$, however, real-time sensing is limited due to the time required to rotate the

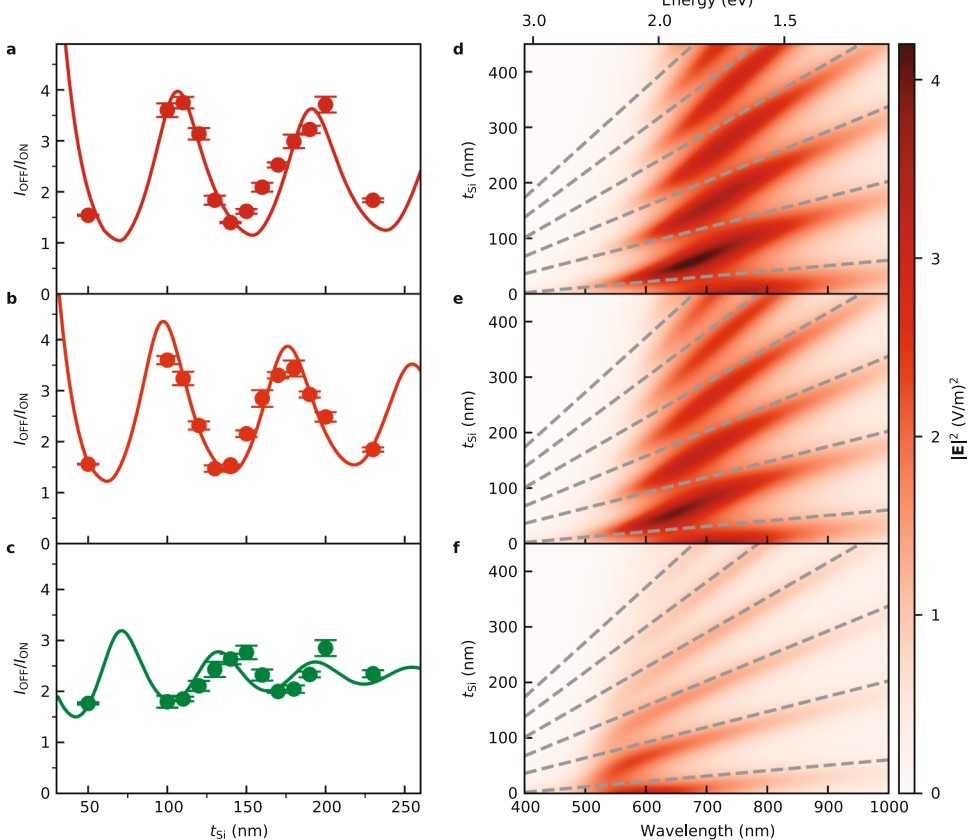

**Fig. 2 Enhancement of detection and coupling to FP modes. a–c** Ratio of the OFF resonance current (at the fixed angle of 25°) and the ON resonance current (at the SPR angle) as a function of silicon thickness for three photon wavelengths (670 nm, 635 nm, 532 nm, respectively) with gold thickness of 30 nm. Symbols with error bars are the mean and SD of four samples. Solid lines are the simulated results with numbers indicating the cavity modes. **d–f** Simulated near field electric field intensity at the gold/air interface as a function of silicon layer thickness and wavelength (photon energy) at 28.1°, 28.5°, and 31.1°, respectively. Dashed lines are linear best fits to the FP modes in the absence of SPR. Source data are provided as a Source Data file.

prism. To highlight the role of the cavity the figure contains two curves with different silicon thicknesses—one able and one unable to support cavity modes. For both devices the current is smallest at the SPR angle due to the reduced reflection by the gold layer. The thicker device shows a moderate increase in photocurrent at other angles due to the light reflected back through the silicon layer. The thinner device shows a much larger increase in photocurrent due to multiple reflections within the cavity.

The angular dependence can be further understood by looking at the electric field intensity, $|\mathbf{E}|^2$, as a function of angle and distance through the device as simulated using Finite-Difference Time-Domain (FDTD) software (Fig. 1d). The silicon layer thickness in this figure (110 nm) corresponds to the blue line in Fig. 1c. At shallow and steep angles regions of high electric field intensity are observed at both edges of the silicon layer ($0 < x < 25$ and $75 < x < 110$ nm) corresponding to the two supported cavity modes – data for other cavity modes can be found in Supplementary Note 3. The high electric field intensity in these regions results in high absorbance and large photocurrent as described previously. At $\theta_{rot} = 28.1°$, a plasmon is formed as seen by the large electric field intensity at the upper surface of the gold layer between $x = 125$ and $140$ nm. Correspondingly, there is a significant decrease in the electric field intensity in the silicon layer at this angle due to the reduced reflection from the gold layer. There is a clear correspondence between the minimum in experimental photocurrent in Fig. 1c and the simulated plasmon field in Fig. 1d.

**Cavity Enhancement & Mode Coupling**. The cavity enhancement is quantified using the ratio of the current with and without plasmon excitation. In Fig. 2a-c we define $I_{OFF/ON}$ as the ratio of the current at the fixed angle 25° (OFF resonance) to the minimum current (ON resonance) at three different wavelengths (670, 635, and 532 nm respectively). The FP modes are evident from the clear wavelength dependent oscillatory enhancement of $I_{OFF/ON}$ as a function of silicon thickness. The observed OFF/ON ratio has peak values up to a factor of 4 at integer values of the cavity mode number $N$ in close agreement with the simulations (see Methods). Simulations predict that the first cavity mode would have the greatest coupling to the plasmon, however, for practical experimental reasons we were unable to fabricate working diodes with such thin silicon layers due to high leakage current. Additionally a compromise has to be made between the higher $I_{OFF/ON}$ ratio of thin layers and the large current possible with thicker silicon layers.

The enhancement of the current at resonant silicon thickness is accompanied by coupling of the plasmon and FP modes. The mode coupling is evident experimentally from small red-shifts of $\theta_{SPR}$ up to a maximum of 0.5° in the studied range (see Fig. 1c). Mode coupling is further illustrated in Fig. 2d–f. Here the simulated near field $|\mathbf{E}|^2$ at the gold/air interface is plotted as a function of wavelength and silicon thickness at fixed angles of illumination. The three plotted fixed angles, $\theta_{rot} = 28.1°$, 28.5° and 31.1°, correspond to $\theta_{SPR}$ found experimentally at the three wavelengths in Fig. 2a–c. The single plasmon modes are seen as broad regions of higher $|\mathbf{E}|^2$ centered at the corresponding

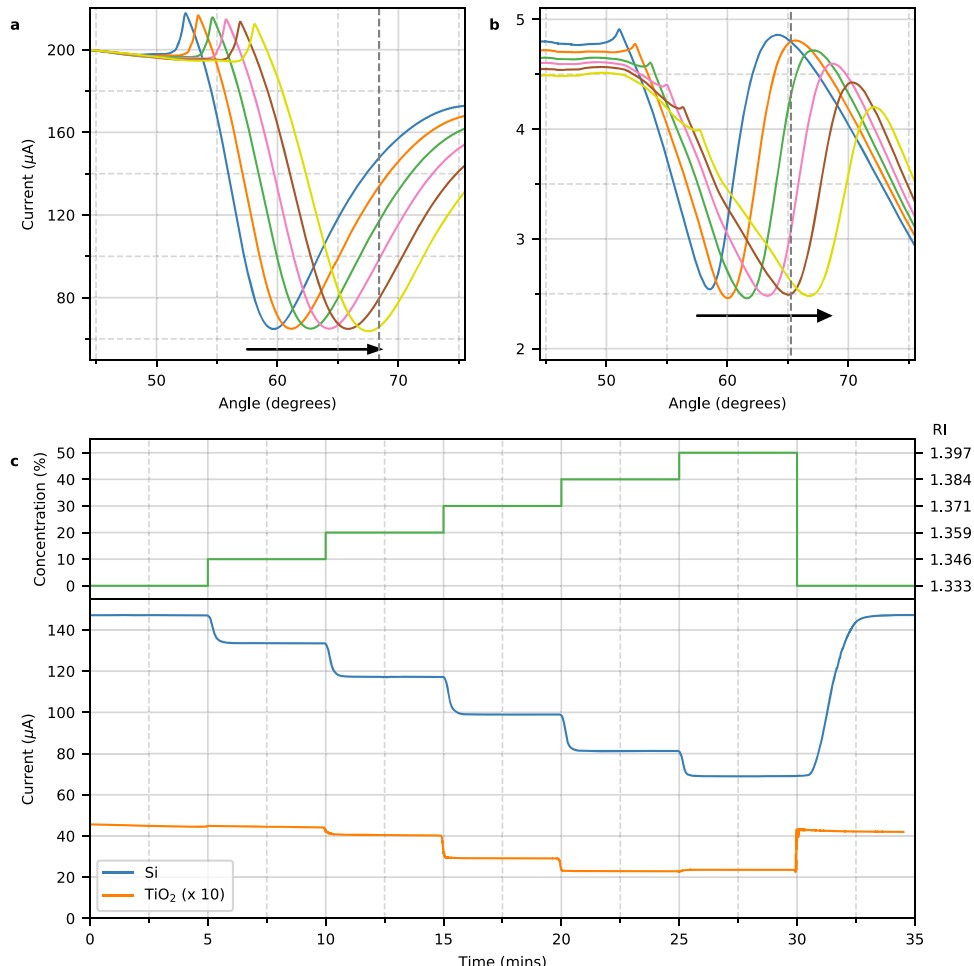

**Fig. 3 Refractive Index sensing. a, b** Photocurrent as a function of angle in solutions of different glycerol concentration [0%, 10%, 20%, 30%, 40%, 50%] with RI = 1.333, 1.346, 1.359, 1.371, 1.384, 1.397 for silicon sample **a** and $TiO_2$ sample **b. c** Photocurrent as a function of time for changing glycerol concentration at the fixed angles indicated by the dashed lines in **a**, **b** for Si (blue) and $TiO_2$ (orange). The $TiO_2$ data is multiplied by a factor of 10 for clarity. The upper panel (green) indicates the corresponding glycerol concentration (left axis) and RI (right axis) of the solution. Source data are provided as a Source Data file.

wavelengths of Fig. 2a–c (670, 635, and 532 nm). Mode coupling is seen as the divergence from the FP modes found in the absence of SPR as determined from simulations at the non-resonant angle of 25° (dashed lines).

**Demonstration of refractive index sensing**. Next we turn to the applicability of the device to detect the molecular interaction between a ligand and an analyte, such as the antigen–antibody interaction discussed in the opening figure. Just as with conventional reflection based SPR sensors, the SPR angle depends on the refractive index on the surface of the gold layer. We use solutions of different glycerol concentration to calibrate the RI dependence of the device due to the near linear dependence of RI with concentration. Figure 3a shows clear linear shifts of the SPR angle of 0.13 degrees/mRIU over the full range of RI.

To show further generality of the electric measurement technique we repeat the experiment with a different type of structure based on hot electron transport. In this second device the silicon layer is replaced with a transparent $TiO_2$ layer. A thicker gold layer (50 nm) is used in this device in order to increase adhesion to the $TiO_2$ layer and prevent peeling. The angular dependence of the electric field intensity plot of this type of device (see Supplementary Note 4) closely resembles that of the silicon-based device of Fig. 1c, however, in this case there is no

direct photon absorption within the semiconductor layer due to the optical transparency of $TiO_2$. Instead, energetic hot electrons are photoexcited inside the gold layer[24–26] that are then emitted over the Schottky barrier into the semiconductor[27] in a three step process[28,29]. Once more the formation of the surface plasmon polariton affects the current, however, in this case this is due to the change of the absorption within the gold layer itself. On resonance the total hot electron generation rate increases, but, the electron distribution moves away from the semiconductor/metal interface to the opposite side of the metal film. As a result the emission rate of hot electrons into the semiconductor decreases and so does the photocurrent. Despite the different current generation mechanisms, the angular and RI dependence of the current is similar for both hot electron transport ($TiO_2$) and direct photogeneration (Si) based devices.

In order to illustrate the real time operation of the devices we fix the angle and flow solutions containing differing glycerol concentrations through the sample chamber. To give more linear RI dependence, the fixed angle was set to $\theta_{rot} > \theta_{SPR}$ as indicated by the dashed lines in Fig. 3a, b. The chamber is initially filled with water and every five minutes the chamber is flushed with a higher concentration of glycerol until finally the chamber is flushed with water once more. The upper panel of Fig. 3c indicates the glycerol concentration and RI as a function of time

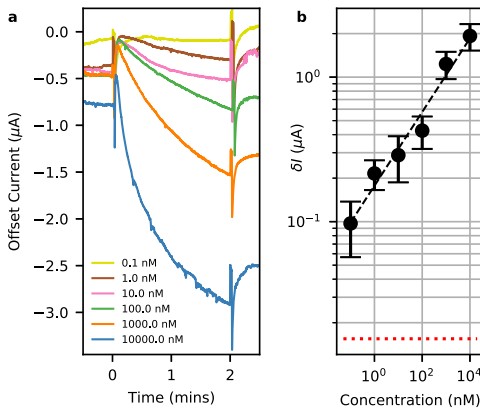

**Fig. 4 Detection of MBP-CoV-2NP. a** Offset photocurrent at fixed angle as a function of time. The sensor is initially covered with a buffer solution. At time = 0, the buffer solution is removed and replaced with solutions containing different concentrations of antigen (MBP-CoV-2NP). At time = 2 min the antigen solution is removed and replaced with buffer solution. **b** Change of photocurrent after 1 min as a function of antigen concentration. Symbols and error bars are the mean and SD of four different sensors. The dotted red line is the noise floor of the sensor as defined by the standard deviation of the current during the initial buffer solution stage (time = −0.5 to 0 min). The dashed black line is a best-fit guide to the eye. Source data are provided as a Source Data file.

and the lower panel shows the corresponding photocurrent of the two types of device. Clear response is seen for both types of device with the slower response time of the silicon device being due to reduced pumping speed. The direct photogeneration based device has a sensitivity of 1.5 μA/mRIU over the entire range of RI (64 mRIU). The hot electron based device has a peak sensitivity of 0.1 μA/mRIU over a narrower range of RI (38 mRIU). (Real time operation of a prototype device can be found in Supplementary Movie.)

**Sensing of antigen–antibody interaction**. Finally, to illustrate the potential of our device for time-dependent molecular sensing, we show in Fig. 4 detection of the MBP (Maltose-Binding Protein)-fused SARS-CoV-2 nucleocapsid protein (MBP-CoV-2NP) over five orders of magnitude of concentration down to 0.1 nM. The experiment was performed in fixed angle mode similar to Fig. 3c. Time dependence measurements in fixed angle mode of operation allow for simple confirmation that the change of refractive index detected by the sensor is due to antigen–antibody interactions and not simple fluctuations in the bulk RI of the different solutions used in the experiment (see Supplementary Note 6). The surface of the gold layer is coated with an antibody via CMD (Carboxymethyl dextran) by amine coupling method. The interaction of the antibody with solutions containing antigens at different levels of concentration was detected using the following manually timed procedure. At first the gold layer is covered with an inert buffer solution, then, at time = 0 the buffer solution is removed and replaced with the antigen containing solution. At time = 2 min the antigen containing solution is removed and replaced with the inert buffer solution. The experiment is repeated from low to high concentrations of antigen with the antibody regenerated between experiments.

Between time = 0 and 2 min the current decreases logarithmically due to a local change of refractive index caused by the interaction of antigens in the solution and antibodies bound to the surface of the gold layer. By continuously monitoring the current in the fixed angle mode of operation, not only can we confirm that the change is due to antigen–antibody interaction,

but we can also determine the affinity, concentration, and binding kinetics of this interaction. The magnitude and rate of the current change depends on the concentration of the antigen; high concentrations show bigger and faster changes of the current. To quantify the interaction strength we plot in Fig. 4b the change in current, $\delta I$, at time = 1 min after first offsetting the data by fitting the current with a logarithmic function to compensate for the abrupt change caused by the exchange of solutions at time = 0. Figure 4b shows the average and standard deviation of $\delta I$ of four samples as a function of molar concentration with the horizontal dashed line indicating the noise floor of the measurement.

## Discussion

We compare the performance of our sensor to that of the commonly used biosensor, BLItz. Using the same antigen and antibody solutions we found the BLItz to saturate at both high and low concentrations and to have a lower detection limit of between 1 and 10 nM (see Supplementary Note 5). Our sensor is sensitive to concentrations at least an order of magnitude lower than the BLItz with no indication of saturation and still well above the measured noise floor. Furthermore, the concentration of nucleocapsid protein in serum to diagnosis SARS-CoV-2 infected patients at early stage is 10 pg/ml (correspond to 250 fM)[30]. According to the best fit in Fig. 4b such low concentration would require $\delta I$ ~20 nA—close to the noise floor of our device. Improvements to the electronics and better temperature & laser power control will improve sensitivity to this level, however, we caution that reducing the signal noise floor on its own may not be sufficient because the device may suffer from saturation effects at small concentrations in a similar manner to the BLItz device.

The efficiencies of the devices are best quantified by the incident photon to conducting electron factor (IPCE). The best IPCE at 670 nm is 10.4% for the silicon-based device and 0.22% for the TiO₂-based device. The two parameters limiting the efficiency of the silicon-based type are the photon absorptance and electron recombination/scattering. The latter may be improved through common electrical engineering techniques, however, care must be taken when attempting to improve the absorptance as this may detrimentally reduce the contrast between OFF- and ON-resonant current. The lower efficiency of the TiO₂ based device is due to the three step hot electron transport mechanism described above. The internal photoemission efficiency $\eta$ given by Fowler[28] is

$$\eta = \frac{(\hbar\omega - \phi_b)^2}{4E_F\hbar\omega} \quad (1)$$

where $\hbar$ is the reduced Planck constant and $\omega$ is the photon frequency. Using a Fermi energy of gold $E_F = 5.53$ eV[31] and a Schottky barrier height for gold and TiO₂ of $\phi_b = 1.0$ eV[32] gives $\eta = 1.8\%$ in reasonable agreement with the difference between the IPCE for the two types of device.

The cavity resonances of Fig. 2 indicate a relatively small finesse of approximately 3. While higher finesse is preferable in fixed-angle mode of operation due to increased $I_{OFF/ON}$ sensitivity, the smaller finesse employed in this work—coupled with the large refractive index of silicon ($n = 4.089$)—ensures that FP modes are conserved over the wide range of angles studied. The finesse may be increased by reducing current generation at the SPR angle, for example by replacing the silicon layer with less absorbing material or operating at less absorbing wavelengths, and/or increasing current generation at non-SPR angles, for example by inserting a thin reflective layer between the silicon and ITO layers to increase cavity confinement.

We have demonstrated a type of silicon photodiode for sensing biomolecular interactions that exploits the physics of surface

plasmons and cavities. Visible light is used to generate a current in an amorphous silicon layer driven by the in-built potential of a Schottky barrier formed at the semiconductor-metal interface. The current is shown to significantly reduce at the critical angle on the formation of the SPR. The reduction of the current is ascribed to the reduced reflection of the metal layer at the critical angle consequently reducing the absorption within the silicon layer as confirmed by simulations. Furthermore, the detection mechanism is enhanced by selecting the silicon thickness such that the SPR is coupled to the FP modes of the absorbing nanocavity. The resulting angular dependence of the photo-current is applied with high contrast to RI sensing in the mold of SPR reflection methods. Using these techniques we were able to detect the protein–protein interaction.

The presented electrical detection scheme used in combination with microfluidics and LED light sources, may greatly reduce the cost and size of plasmon based molecular sensors. In addition the device design lends itself to micropatterning[33,34] that may enable direct detection of local changes in the refractive index for use in the study of cell morphology. The combination of plasmonic and photonic cavity technologies may open a new realm of sensing and current generation technologies.

## Methods

**Device fabrication**. The substrate from GEOMATEC Co., Ltd. consists of 1.1 mm thick high refractive index glass ($n = 1.7746$, S-TIH11, OHARA INC.) covered by a thin transparent electrode of Indium-Tin-Oxide. Semiconductor layers (either amorphous silicon or TiO$_2$) were deposited using magnetron sputtering. A 30 or 50 nm gold layer was then deposited by magnetron sputtering. An electrode was connected to the Au layer by silver paint and the device mounted onto a high refractive index prism ($n = 1.7746$, S-TIH11, OHARA INC.) using diiodomethane ($n = 1.737$, FUJIFILM Wako Pure Chemical Corporation) as a refractive index matching layer. The prism was mounted into a home-made housing fixed onto a rotating stage and the device connected to an ALS CH Instruments Electrochemical Analyzer 802D or Keysight Technologies B1500A for electrical measurements. The prism was illuminated by P-polarized light from a Thorlabs CPS670F continuous wave laser of wavelength 670 nm at incident power of 4 mW with a spot size of diameter approximately 1 mm. The reflected light power was optionally measured using a Newport 843-R power meter. The refractive indices of the solutions with different glycerol concentrations were measured using an Atago PAL-RI refract-ometer and pumped through the housing.

**Antibody and antigen preparation**. Anti-SARS-CoV-2 nucleocapsid protein antibody [1A6] (ab272852) was purchased from abcam. As antigen, MBP (Maltose-Binding Protein)-fused SARS-CoV-2 nucleocapsid protein (MBP-CoV-2NP) was purified as the following. An expression vector, pET28a was purchased from Merck Millipore and deleted T7-tag region by invert PCR (designated pET28aΔT7)[35]. MBP-CoV-2NP was constructed by inserting MBP gene from pMAL-c5x (from New England Biolabs Inc.), and then, the SARS-CoV-2 nucleocapsid gene (from Integrated DNA Technologies, Inc. as an E. coli-codon-optimized synthetic gene of Uniprot ID: P0DTC9) into BamHI/EcoRI and EcoRI/HindIII sites of pET28aΔT7, respectively. MBP-CoV-2NP was expressed in E. coli BL21(DE3) and purified Ni Sepharose High Performance (cytiva, formerly GE healthcare) using 50 mM Tris-HCl pH 9, 1 M NaCl, 0.01% TritonX-100 as a purification buffer.

**Protein–protein interaction**. To immobilize antibody on a sensorchips, Carboxymethyl dextran layer (500 kDa, CMD-500, Meito sangyo) was formed via 11-AUT (11-Amino-1-undecanethiol hydrochloride, DOJINDO). After forming the CMD layer, the device was mounted on the measurement system mentioned above. Antibody was immobilized using amine coupling method by dropping the mixture of 50 mM sulfo-NHS (N-hydroxysuccinimide, FUJIFILM Wako Pure Chemical Corporation) and 200 mM WSC (water-soluble carbodiimide, DOJINDO), sulfo-NHS/WSC at Fig. 4, and then, dropped 100 μM anti-SARS-CoV-2 nucleocapsid protein antibody diluted in 10 mM MES pH 6.0 onto the surface of the device. To confirm whether antibody was immobilized or not, 10 mM MES pH 6.0 (Buffer) was dropped onto a sensorchip before and after the immobilization procedure. For the measurement of protein–protein interaction, the chamber was filled with running buffer (50 mM Tris-HCl pH 9, 1 M NaCl, 0.01% TritonX-100) to measure the baseline level (30 s). The association and dissociation were measured by dropping various concentration of MBP-CoV-2NP protein solutions and running buffer for 120 s each, respectively. After each measurement, a sensor surface was regenerated by 100 mM Glycine pH 2 for 30 s three times. Comparison of the sensitivity between our sensor and a commercially available sensor was performed using BLItz (Sartorius) and AR2G sensorchips by the same procedure as our sensor.

**Optical and electrical simulation**. All simulations were performed using "FDTD Solutions" from Lumerical Inc. with the results obtained using an analytical transfer matrix method for 1D simulations (functions "stackrt" and "stackfield") or a full finite-difference time-domain solver for 3D simulations. The simulation details are given in the below table for 1D simulations. For 3D simulations the glass layer was removed and a coarser spatial resolution was used due to memory and processing time constraints. Data analysis was performed and plotted using Python packages numpy and matplotlib.

| Layer | Thickness (nm) | Refractive Index at $\lambda = 670$ nm | Extinction Coefficient at $\lambda = 670$ nm |
|---|---|---|---|
| Glass | Infinite | 1.7746 | 0 |
| TiO$_2$ | 200 | 2.42 | 0 |
| Amorphous silicon | variable | 4.089 | 0.114 |
| Au | 30 or 50 | 0.335 | 3.802 |
| Surface dielectric | Infinite | 1.0–1.4 | 0 |

**Typical 1D simulation parameters**. Optical properties of Au were taken from the "CRC Handbook of Chemistry & Physics"[36]. Other optical properties were determined experimentally.

The reflectance ($R$), transmittance ($T$), and electric field intensity ($|\mathbf{E}|^2$) were calculated directly. The total absorptance ($A$) was calculated from $A = 1-T-R$ or by integrating the $|\mathbf{E}|^2$ over spatial coordinates using

$$A = -\frac{1}{2} \iiint \frac{\omega \varepsilon_i |E(x,y,z)|^2}{P} \, dz \, dy \, dx \quad (2)$$

where $\omega$ is the angular frequency of the incident light, $\varepsilon_i$ is the imaginary part of the permittivity, and $P$ is the incident power. To mimic the photocurrent we introduce the empirical quantity *contributing absorptance*, $CA$, given by

$$CA = -\frac{1}{2} \int_{x=0}^{\Delta} \iint \frac{\omega \varepsilon_i |E(x,y,z)|^2}{P} \, dz \, dy \, dx \quad (3)$$

where the integral depends on the type of semiconductor being used. For the amorphous silicon based device the current is predominantly generated within the silicon layer near the silicon/gold interface. For the TiO$_2$ based device the current is exclusively generated within the gold layer near the TiO$_2$/gold interface. The integration thickness $\Delta$ is treated as a fitting parameter, typically 20–60% of the silicon layer thickness (for the silicon based device) or ~20% of the gold thickness (for the TiO$_2$ based device).

As a final step, $R$, $T$, $|\mathbf{E}|^2$, and $CA$ were converted to the experimental frame by applying Snell's Law and simple trigonometric transformations given in Supplementary Note 1.

**Reporting summary**. Further information on research design is available in the Nature Research Reporting Summary linked to this article.

## Data availability
Source data are provided with this paper.

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

## Acknowledgements
We thank N. Hayakawa, N. Gorai and M. Ando for fruitful discussions and W. Hasebe, A. Miyaji and M. Sasaki for technical assistance. HM acknowledges the financial support from JSPS KAKENHI (Grants JP18H05205). This work was supported by IMRA Japan Co., Ltd.

## Author contributions
H.S. and K.H. designed and supervised the project. G.A., A.K.S., Y.O., H.K., K.H. and H.S. performed experiments and analyzed data. G.A. performed simulations and jointly conceived the model for contributing absorptance with H.S. K.U. and H.M. gave support and conceptual advice. G.A. wrote the paper with comments from all coauthors.

## Competing interests
G.A., A.K.S., Y.O., H.K., K.H. and H.S. work for IMRA Japan Co., Ltd. IMRA Japan Co., Ltd. and Hokkaido University applied a patent for the structure and function of TiO$_2$-based surface plasmon resonance sensor, with H.S., G.A., M.S., K.H., H.M. and K.U. as inventors. The PCT Application no. is WO2019031591A1 and the applied patent was transited to United States (US10983052, patented), Europe (EP3667297A), China (CN110199191A), and Japan (WO19/031591). IMRA Japan Co., Ltd. applied a patent for the structure and function of a Fabry-Pérot cavity coupled surface plasmon sensor, with A.K.S., G.A., Le Viet Cuong, H.S., H.K., M.A. as inventors (PCT Application no.: WO2021075529A1 under transition to US, EP, CN, and JP). There are no other competing interests.
