## [Peer Review File · Nature Communications]

Reviewers' comments:

Reviewer #1 (Remarks to the Author):

The paper describes simulations and measurements of surface plasmon resonance using a new idea which exploits a silicon Fabry-Perot (finesse ≈ 3) as both the sensor and as a signal enhancing element. Furthermore the signal is extracted as a current from the silicon photodiode simplifying the measurement scheme. A brief comparison is also made with a TiO₂ layer instead of Si.

I think the paper should be published after the corrections listed below. I have divided the corrections in two parts: Figures and text.

Please define the acronyms: RI on line 73, SPP on line 108, RIU on line 134, TIR on line 169. Also define the electric field intensity as $|E|^2$ on line 84.

Figures:

In general Figure 1a) b) c) d) and e) are too small.

1) In Figures 1a) and 1b) it is not clear with respect to which surface is the incident angle.

Furthermore what are the angles of the prism. The beam of light does not go in a straight line through the glass and silicon elements and should be indicated (roughly). In lines 81 and 82 an incident angle range and a refracted angle range in silicon are reported which cannot be reproduced by the reader without this information.

2) The gold thickness should be indicated (from $x = 100$ nm to $x = 150$ nm I understand) and a comment that the field intensity at 28.5 degrees and 125 nm $< x < 150$ nm (in the gold) is the plasmon field. This helps understand figure 1d). What wavelength is simulated? Figure 1e) shows ≈ 2 eV $\Rightarrow \approx 623$ nm whereas 670 nm $\Rightarrow 1.86$ eV.

3) It would be useful to also have the wavelength indicated on the x axis of Figure 1d) rather than just the photon energy. Or maybe graph as a function of the wavelength rather than the energy. Comment on the shape the dotted curve should have in the absence of the Fabry-Perot coupling (proportional to $1/E$ or λ).

4) In Figure 2 results are also given for a 30 nm thick gold layer but nothing is mentioned in the text. Furthermore all other figures refer to a gold thickness of 50 nm. Comment this thickness with respect to the 50 nm thickness or eliminate. The impression is that the 30 nm thickness is reported only because it seems to improve the peak to valley ratio with respect to the 50 nm but all results are given with the 50 nm thickness. Explain.

5) In Figure 3c) also indicate the index of refraction variation together with the glycerol concentration. Furthermore on line 133 - 134 you report a calibration of 0.11 degrees/mRIU which can in no way be reconstructed by the reader. It would also be interesting to determine the standard deviation of the current at the various glycerol concentrations so as to estimate a minimum RIU sensitivity.

6) Figure S2 reports simulations at different wavelengths without any comment. Indicate in the caption the gold thickness (50 nm).

7) for a comparison with the other simulations maybe in Figure S3 a linear scale should be used for the field intensity.

Text:

Line 66: It would be interesting to have the finesse of the Fabry-Perot written somewhere in the text.

Line 74: Indicate with respect to which surface is the incident angle defined.

Line 80: I don't understand the word 'vitality'.

Line 81: Either refer to the table on line 247 or indicate the index of refraction of Silicon and glass.

Line 82: Again, the reader must be able to reconstruct the angular ranges indicated using the various angles and indices of refraction.

Lines 86-92: Referring to Figure 1c), indicate in the text the gold layer coordinates (and thickness) so as to make it clear that the field intensity at 125 nm $< x < 150$ nm is the plasmon field.

Line 97: the four-fold amplification is only obtained with a gold layer of 30 nm. In no other measurement/simulation in the paper is this thickness used. What is the drawback of this 30 nm thickness?

Line 102: There is indeed a current dip enhancement due to the Fabry-Perot but I believe that this enhancement is mainly due to the off-SPR current increase (stronger field intensity inside the silicon). Indeed in Figure 1d) the current increase off-SPR is more than a factor 3. There is a coupling between the Fabry-Perot and the plasmon field (Figure 1e)) but it seems to me that the dip in current at the SPR angle is mainly due to the decrease in reflectivity at the silicon/gold interface killing the Fabry-Perot modes.

Line 105: indicate the wavelength in Figure 1e).

Line 107: deviates from what? Indicate the expected energy-thickness relation (or wavelength-thickness).

Line 108: In Figure 1d) it is very difficult to evaluate the shift of 0.5 degrees reported in the text. It is not evident. Again a larger figure with maybe some dotted vertical lines or something similar.

Line 119: Again the factor 4 gain is only true for a gold thickness of 30 nm which doesn't seem to be used anywhere else in the paper.

Line 133: the 0.11 degrees/mRIU cannot be extracted by the reader in any way. Add information (as mentioned above)

Line 207: the value of $n = 1.7746$ is reported as 1.7744 in the table.

Line 215: P-polarized with a capital 'P'

Line 217: more correct is the use of 'indices'

Lines 250-256: reflectance, transmittance and absorptance are adimensional quantities. The integral expressions have units of power. Please correct.

Reviewer #2 (Remarks to the Author):

The presented work is interesting and demonstrates the value of resonantly enhanced light-matter interaction. The manuscript claims that plasmonic sensing, together with the Fabry-Perot cavity resonance in a photodetector, could simplify the experimental setup. The authors, however, do not compare the sensitivity of the presented approach with the conventional angle sensing approach. In the case of angle sensing, the shape of the reflectance curve vs. angle is sufficient to sense the analyte concentration. An intensity calibration is not necessary. Also, when relying upon the shape of a curve, only the relative angle shift is sufficient to sense the analyte. Plus, the shift in peak can be detected more accurately by curve fitting techniques. However, the presented approach converts the angular to intensity sensitivity. Intensity-based measurement requires calibration. Plus, it is a single point measurement and hence do not take advantage of curve fitting or peak tracking. Hence, the presented approach is very similar to the traditional non-plasmonic sensing approaches.

The authors motivate their work as a compact sensor by eliminating the need for an angular sweep. However, their protein-protein interaction demonstration makes use of angular sweep. Again here, if the angular sweep is carried out, the presented work needs to be compared against a control case (traditional SPP sensor).

Reviewer #3 (Remarks to the Author):

The manuscript presents an optoelectronic approach to propagating surface plasmon sensors by depositing the metal film on a silicon Fabry-Perot cavity. The generated photocurrent is sensitive to the refractive index of the environment (a solution).

Overall, this is a short manuscript with an eminently applied character. I have doubts about its significance for the field of SPR sensors because the authors have not quantified what makes this device better or worse compared to the state of the art with hard numbers. The concept is simple, which could be good for applications, but I cannot expect it to influence further developments in the field.

- What is the actual benefit of this optoelectronic device compared to a conventional SPR sensor? Only integration? The authors briefly mention size, cost, and calibration, but they are not quantifiable in the manuscript. On the other hand, how does performance compare in terms of other quantities such as lowest detectable refractive index changes? Specifically, it seems to me that the ON/OFF ratio of this device is substantially lower than the equivalent ratio of a conventional SPR sensor.
- Can the authors describe the impact of the higher refractive index and optical losses of the silicon layer on the top PSRP sensor and its sensitivity compared to more conventional dielectric substrates for the metal film?
- Please elaborate on available evidence and the meaning of the following sentence: "with the hot electron-based device having a smaller working range of RI than the direct photogeneration based device with the latter varying monotonically over the entire range of RI". From Figure 3c, I just see a lower photocurrent for the TiO₂ device, but the monotonically decreasing current occurs for both devices. The exact dependence on concentration depends on the angle chosen, but the shapes of figures 3a and 3b are qualitatively similar to a first approximation.
- The work is succinct and simple to understand but unnecessarily difficult to read, in part because of lack of information in figures and their captions. For example, in Figure 2, the meaning of the ON/OFF ration could be introduced in more detail for clarity. Also in Figure 2, the wavelengths and thickness of the dashed line layer are not described. Another example: a legend in Figure 3c could save time to the reader by indicating which device it corresponds to.
- Indicate the meaning of the dashed line in Figure 1c, as the origin of the x-axis and the thickness of the metal film are not described in the caption.

Response to reviewers

On behalf of all authors I wish to express our deep gratitude to the reviewers for their extremely useful comments regarding our manuscript. As a result of these comments we have performed new experiments and made wholesale changes to the manuscript, including changing all figures. We believe that these changes improve the quality of the work, quantify the performance of our device, and put it into context by comparing with the state of the art.

Before addressing each of the reviewer comments in turn, we wish to begin by giving a brief overview of the three major changes to the manuscript and the new experiments that were involved.

1. The results and discussion sections have been substantially re-written and restructured in order to improve readability and to clarify comments made by the reviewers. All figures have been changed.
2. We have removed all reference to the device with gold thickness of 50 nm and replaced with data for gold thickness of 30 nm.
3. We have changed the target protein interaction in Fig. 4 to the MBP-fused SARS-CoV-2 nucleocapsid protein (MBP-CoV-2NP). Experiments are now performed in fixed angle mode of operation over five orders of magnitude of concentration. Results are compared to a commonly used biosensor and we note that the sensitivity of our device may be sufficient, in principle, to diagnose SARS-CoV-2 infected patients at early stage.

Here follows our point-by-point response to the comments from the reviewers. Remarks from reviewers are copied in italic font and our responses are given in roman font.

Reviewer #1 (Remarks to the Author):

The paper describes simulations and measurements of surface plasmon resonance using a new idea which exploits a silicon Fabry-Perot (finesse ≈ 3) as both the sensor and as a signal enhancing element. Furthermore the signal is extracted as a current from the silicon photodiode simplifying the measurement scheme. A brief comparison is also made with a TiO₂ layer instead of Si.

I think the paper should be published after the corrections listed below. I have divided the corrections in two parts: Figures and text.

We thank the reviewer greatly appreciate their comments that have enabled several improvements to the manuscript.

Please define the acronyms: RI on line 73, SPP on line 108, RIU on line 134, TIR on line 169. Also define the electric field intensity as $|E|^2$ on line 84.

These changes have been made where they remain in the new version of the manuscript.

Figures:

In general Figure 1a) b) c) d) and e) are too small.

Figure 1e) has been moved to Fig. 2 in the restructuring of the paper. As a result the remaining figures are now bigger and should be more readable.

1) In Figures 1a) and 1b) it is not clear with respect to which surface is the incident angle. Furthermore what are the angles of the prism. The beam of light does not go in a straight line through the glass and silicon elements and should be indicated (roughly). In lines 81 and 82 an incident angle range and a refracted angle range in silicon are reported which cannot be reproduced by the reader without this information.

We thank the reviewer for pointing out the lack of clarity in the figure. To make the figure clearer we have added a small inset to indicate the definition of the experimental rotation angle θ_{rot} . We have changed the path of the beam of light to illustrate that it does not continue in a straight line through the glass and silicon layers, but is bent due to the difference in refractive index. Furthermore, we have added a section to the supplementary information ("Angle Conversion") within which we give the formulae for converting between incident angle θ_{rot} and other relevant angles.

2) The gold thickness should be indicated (from $x = 100$ nm to $x = 150$ nm I understand) and a comment that the field intensity at 28.5 degrees and 125 nm $< x < 150$ nm (in the gold) is the plasmon field. This helps understand figure 1d). What wavelength is simulated? Figure 1e) shows ≈ 2 eV $\Rightarrow \approx 623$ nm whereas 670 nm $\Rightarrow 1.86$ eV.

We thank the referee for their comments and now explicitly state the region and angle of the plasmon field. The simulation wavelength for this figure is 670 nm and this information has been added to the figure caption. As discussed above, the data in Fig 1d (formerly Fig 1c) now uses simulations with a gold thickness of 30 nm, however, the qualitative description remains the same.

Regarding the former Fig. 1e, in the latest version we have moved this figure into Fig. 2 and now use data for the 30 nm gold film. Please see the following comment for further information.

3) It would be useful to also have the wavelength indicated on the x axis of Figure 1d) rather than just the photon energy. Or maybe graph as a function of the wavelength rather than the energy. Comment on the shape the dotted curve should have in the absence of the Fabry-Perot coupling (proportional to $1/E$ or λ).

We thank the reviewer for their comment and agree that the presentation of this data should be improved. On further consideration of all reviewers' comments we have made several changes to both the figure and the text. We outline these changes in our response to lines 102, 105, and 107 as shown below.

4) In Figure 2 results are also given for a 30 nm thick gold layer but nothing is mentioned in the text. Furthermore all other figures refer to a gold thickness of 50 nm. Comment this thickness with respect to the 50 nm thickness or eliminate. The impression is that the 30 nm thickness is reported only because it seems to improve the peak to valley ratio with respect to the 50 nm but all results are give with the 50 nm thickness. Explain.

We agree with this comment from the reviewer that the inclusion of data for both 30 nm and 50 nm was not adequately justified. The 50 nm data has been entirely removed and replaced with 30 nm data. At the time of initial submission, 30 nm thick gold was not robust enough to withstand experiments in liquid therefore data for 50 nm thickness was shown in Figs. 3 and 4. After improving fabrication techniques we are now able to use 30 nm thickness for all experiments using the silicon-based sensor.

5) In Figure 3c) also indicate the index of refraction variation together with the glycerol concentration. Furthermore on line 133 - 134 you report a calibration of 0.11 degrees/mRIU wich can in no way be reconstructed by the reader. It would also be interesting to determine the standard deviation of the current at the various glycerol concentrations so as to estimate a minimun RIU sensitivity.

We thank the reviewer for this very useful comment and have added a second axis to this figure to indicate the refractive index. To further enable the calibration to be reconstructed by the reader, we have also added the RI of Fig. 2a and 2b to the figure caption. The standard deviation of the current is 20 nA corresponding to a limit of detection of 13 μ RIU (using the sensitivity of 1.5 μ A/mRIU derived in the manuscript). We now include this information in the "discussion" section of the manuscript in which we compare it with the limit of detection required to diagnose SARS-CoV-2 infected patients at early stage.

6) Figure S2 reports simulations at different wavelengths wihtout any comment. Indicate in the caption the gold thickness (50 nm).

Information about the gold thickness has been added. For consistency with the main text the data has been exchanged with simulations for gold thickness of 30 nm.

7) for a comparison with the other simulations maybe in Figure S3 a linear scale should be used for the field intensity.

We thank the reviewer for pointing out this inconsistency and have changed the figure accordingly.

Text:

We wish to preface the following point-by-point responses by stating that, due to the comprehensive changes to the manuscript, several of these sections have been rewritten entirely. Nonetheless we thank the reviewer for their comments that have helped us to greatly improve the manuscript.

Line 74: Indicate with respect to which surface is the incidente angle defined.

We thank the reviewer for pointing out that this was not previously well described. We clarify this angle by referring to the angle of rotation as shown in the inset of Fig. 1.

Line 80: I don't understand the word 'vitally'.

Line 66: It would be interesting to have the finesse of the Fabry-Perot written somewhere in the text.

We apologise for the poor choice of language. Our intention was to convey that it is essential that the Fabry-Perot mode is conserved while the sample is rotated otherwise the current may change dramatically simply due to loss of the Fabry-Perot enhancement effect. In the discussion section we now explain that the mode is conserved over the complete range of angles due to the high RI of silicon and relatively small finesse of the cavity.

Line 81: Either refer to the table on line 247 or indicate the index of refraction of Silicon and glass.

Line 82: Again, the reader must be able to reconstruct the angular ranges indicated using the various angles and indices of refraction.

Lines 86-92: Refereing to Figuer 1c), indicate in the text the gold layer coordinates (and thickness) so as to make it clear that the field intensity at $125\text{ nm} < x < 150\text{ nm}$ is the plasmon field.

We thank the reviewer for pointing out these missing points. As previously mentioned, the discussion about the large refractive index of silicon that was on Line 81 has been moved to the “discussion” section where we have included its value in the text. The value in the table has also been corrected. As described above, we also include more information to enable the reader to reconstruct the angle ranges. We have clarified the text to make it explicit that the large field intensity (now at $125\text{ nm} < x < 140\text{ nm}$) is the plasmon field in the gold layer.

Line 97: the four-fold amplification is only obtained with a gold layer of 30 nm. In no other measurement/simulation in the paper is this thickness used. What is the drawback of this 30 nm thickness?

We refer to our previous comments regarding the 30 nm and 50 nm films.

Line 102: There is indeed a current dip enhancement due to the Fabry-Perot but I believe that this enhancement is mainly due to the off-SPR current increase (stronger field intensity inside the silicon). Indeed in Figure 1d) the current increase off-SPR is more that a factor 3. There is a coupling between the Fabry-Perot and the plasmon field (Figure 1e)) but it seems to me that the dip in current at the SPR angle is mainly due to the decrease in reflectivity at the silicon/gold interface killing the Fabry-Perot modes.

Line 105: indicate the wavelength in Figure 1e).

Line 107: deviates from what? Indicate the expected energy-thickness realtion (or wavelength-thickness).

We thank the reviewer for these comments along with comment #3 above. On careful consideration we realise that the description of the enhancement of detection and the coupling of the cavity and plasmon modes was not sufficiently explained. We have therefore decided to completely restructure this part of the manuscript and change Figs. 1 and 2 accordingly. As a result, the plot of the simulation data indicating the coupling (former Fig. 1e) has been moved to Fig. 2 where it now acts as a partner to the plot of enhancement of detection.

Before describing the changes to the manuscript, we first wish to describe the expected behaviour in the absence of Fabry-Perot coupling. In Fig. L1 we plot the simulated electric field intensity, $|E|^2$, at the surface of the gold layer using a gold thickness of 30 nm and an angle of incidence $\theta_{rot} = 25^\circ$. This angle is insufficient to generate a plasmon and so the regions of strong electric field are solely due to FP modes. Plotting as a function of wavelength it can be seen that the FP modes depend linearly on the wavelength as indicated by the dashed lines.

Fig. L1: FP modes in the absence of SPR. The dashed line linear guides to the eye are now included in Figs. 2d-f of the main text.

To illustrate the connection between coupling and signal enhancement we consider the pair of figures 2a & 2d in the manuscript. Fig. 2a shows the ratio of the current at $\theta_{rot} = 25^\circ$ (I_{OFF}) and $\theta_{rot} = \theta_{SPR} \sim 28.1^\circ$ (I_{ON}) at a wavelength of 670 nm. Fig. 2d shows the simulated electric field intensity as a function of wavelength and silicon thickness at the fixed angle of $\theta_{rot} = 28.1^\circ$. Fig. 2d resembles Fig. L1 above with the addition of a region of strong intensity centred around 670 nm, i.e. the plasmon wavelength at this angle of incidence. The coupling of the Fabry-Perot and plasmon modes is seen as divergence from the uncoupled case (linear dashed lines) in Figs. 2d.

As noted by the reviewer, the I_{OFF}/I_{ON} ratio is due to the decreased reflectivity at the SPR angle. At silicon thicknesses corresponding to FP modes this ratio is enhanced by increasing the I_{OFF} current due to multiple internal reflections within the cavity. The coupling results in a change of the Fabry-Perot resonance length and a change of SPR angle as discussed in the following comment by the reviewer. Pairs of figures (2a & 2d, 2b & 2e, 2c & 2f) show the consistency between wavelength, angle of incidence, and coupling strength.

Line 108: In Figure 1d) it is very difficult to evaluate the shift of 0.5 degrees reported in the text. It is not evident. Again a larger figure with maybe some dotted vertical lines or something similar.

(As part of the reorganisation of the paper, this figure has been moved to Fig. 1c and the data has been replaced with devices with gold thickness of 30 nm.)

We agree with the referee that the shift of angle was not clear in the former Fig 1d. To increase the visibility of the shift of angle in the new figure we add dotted lines to indicate θ_{SPR} and normalise the current by the value at the SPR angle (I_{OFF}). A further advantage of normalising the data in this way is to make the meaning of the I_{OFF}/I_{ON} ratio used in Fig. 2 more explicit. The I_{OFF}/I_{ON} ratios (~ 4 and ~ 1.3) can be easily determined from this plot and compared with the corresponding thicknesses shown in figure 2a.

Line 119: Again the factor 4 gain is only true for a gold thickness of 30 nm which doesn't seem to be used anywhere else in the paper.

Please see the previous comments regarding the gold layer thickness.

Line 133: the 0.11 degrees/mRIU cannot be extracted by the reader in any way. Add information (as mentioned above)

We apologise for this oversight and include the information necessary for the reader.

Line 207: the value of $n = 1.7746$ is reported as 1.7744 in the table.

Line 215: P-polarized with a capital 'P'

Line 217: more correct is the use of 'indices'

Lines 250-256: reflectance, transmittance and absorptance are adimensional quantities. The integral expressions have units of power. Please correct.

We thank the reviewer for pointing out these errors that have now been corrected.

Reviewer #2 (Remarks to the Author [numbering added for clarity]):

2-1) The presented work is interesting and demonstrates the value of resonantly enhanced light-matter interaction. The manuscript claims that plasmonic sensing, together with the Fabry-Perot cavity resonance in a photodetector, could simplify the experimental setup.

We thank the reviewer for noting that our device may simplify the experimental set-up for plasmonic sensing.

2-2) The authors, however, do not compare the sensitivity of the presented approach with the conventional angle sensing approach. In the case of angle sensing, the shape of the reflectance curve vs. angle is sufficient to sense the analyte concentration. An intensity calibration is not necessary. Also, when relying upon the shape of a curve, only the relative angle shift is sufficient to sense the analyte. Plus, the shift in peak can be detected more accurately by curve fitting techniques. However, the presented approach converts the angular to intensity sensitivity. Intensity-based measurement requires calibration. Plus, it is a single point measurement and hence do not take advantage of curve fitting or peak tracking. Hence, the presented approach is very similar to the traditional non-plasmonic sensing approaches.

The authors motivate their work as a compact sensor by eliminating the need for an angular sweep. However, their protein-protein interaction demonstration makes use of angular sweep. Again here, if the angular sweep is carried out, the presented work needs to be compared against a control case (traditional SPP sensor).

We thank the reviewer for their comments describing the limitation of the presented work. These comments have proven to be extremely useful to improve the quality of the work, quantify the performance of our device, and put it into context by comparing with the state of the art. We now describe our efforts to address the reviewer's reservations.

Firstly, as noted by the reviewer, in the former Fig. 4 we demonstrated protein-protein interaction by making use of the angular sweep, thus making no use of the fixed angle mode of operation that we promoted in the earlier part of the manuscript. For our purpose, along with the simplicity of operation, the main advantage of the fixed angle mode of operation is the ease at which the time dependence, and thus the binding kinetics of the interaction, can be extracted. This inconsistency in the manuscript has now been rectified by replacing Fig. 4 with new fixed angle experiments.

We agree with the reviewer that curve fitting of angle sweep data is an accurate and well-established method of detection, for example Biacore T200 specifies a noise equivalent to 0.000003° . Curve fitting may equally be applied to our sensor, however, we now demonstrate that fixed angle mode is sufficient for our purpose. In the new version of Fig. 4 we use the fixed angle mode of operation to detect in real time the MBP-fused SARS-CoV-2 nucleocapsid protein (MBP-CoV-2NP) over five orders of magnitude of concentration down to 100 pM. We strongly caution that further work is necessary because the signal may saturate at lower concentrations, however, applying the fit to the data we determine that the sensitivity and noise of our sensor are sufficient to detect the concentration required (250 fM) to diagnose SARS-CoV-2 infected patients at early stage [Li]. Using the sensitivities of 1.5 $\mu\text{A}/\text{mRIU}$ and 0.13 $^\circ/\text{mRIU}$ derived from Fig. 3, we determine that the current noise ~ 20 nA corresponds to a detection limit of 13 μRIU equivalent to an angle resolution of 0.002° if operated in the angle sweep operation. We wish to emphasise that the noise level of our sensor is due to technical, not scientific, reasons and may be lowered further with development.

As a further means of comparison, we perform experiments to determine that our sensor outperforms that of the commonly used biosensor, BLitz, in the detection of this protein.

[Li] Li, T., Wang, L., Wang, H., Li, X., Zang, S., Xu Y. & Wei, W. Serum SARS-COV-2 Nucleocapsid Protein: A Sensitivity and Specificity Early Diagnostic Marker for SARS-COV-2 Infection. *Frontiers in Cellular and Infection Microbiology* **10** 470 (2020).

Reviewer #3 (Remarks to the Author [numbering added for clarity]):

3-1) The manuscript presents an optoelectronic approach to propagating surface plasmon sensors by depositing the metal film on a silicon Fabry-Perot cavity. The generated photocurrent is sensitive to the refractive index of the environment (a solution).

Overall, this is a short manuscript with an eminently applied character. I have doubts about its significance for the field of SPR sensors because the authors have not quantified what makes this device better or worse compared to the state of the art with hard numbers. The concept is simple, which could be good for applications, but I cannot expect it to influence further developments in the field.

We thank the reviewer for stating that our concept may be good for applications. We have made several changes to the manuscript in order to quantify the performance of our sensor (see below for a comparison of its performance to that of conventional SPR sensors). In addition, in the latest version of the manuscript we show that our sensor outperforms the widely used BLItz biosensor in the detection of the SARS-CoV-2 virus protein.

3-2) - What is the actual benefit of this optoelectronic device compared to a conventional SPR sensor? Only integration? The authors briefly mention size, cost, and calibration, but they are not quantifiable in the manuscript. On the other hand, how does performance compare in terms of other quantities such as lowest detectable refractive index changes? Specifically, it seems to me that the ON/OFF ratio of this device is substantially lower than the equivalent ratio of a conventional SPR sensor.

We agree with the reviewer that the size, cost, and calibration are not quantifiable benefits in the manuscript. While we feel that these are true benefits for a commercial product, at this stage it is not possible to quantify them, so it is prudent to remove such claims from the manuscript. We shall now discuss the performance with regard to the new data included in Figs. 3 and 4.

In fixed angle mode of operation the ON/OFF ratio by itself does not limit the sensitivity. Instead the more limiting factor is the current noise. Our sensor has noise of 20 nA, which is equivalent to a detection limit of 13 μ RIU (using the sensitivity of 1.5 μ A/mRIU derived in the manuscript). As described in the newest version of the manuscript, this level of detection is, in principle, sufficient to diagnose SARS-CoV-2 infected patients at early stage of 10 pg/ml [Li]. Thus, although the ON/OFF ratio of our device is poorer than a conventional SPR sensor, the current noise is sufficiently small to detect small refractive index changes.

To achieve the same level of detection with angle-dependent reflection measurements of conventional SPR sensors would require an angle resolution of 0.0015° according to simulations. Such angle resolution is certainly achievable – indeed Biacore T200 specifies resolution equivalent to 0.000003° – but the fixed angle mode of operation is much simpler to implement. Furthermore, we wish to emphasise that the noise level of our sensor is due to technical, not scientific, reasons and may be lowered further with development.

[Li] Li, T., Wang, L., Wang, H., Li, X., Zang, S., Xu Y. & Wei, W. Serum SARS-COV-2 Nucleocapsid Protein: A Sensitivity and Specificity Early Diagnostic Marker for SARS-COV-2 Infection. *Frontiers in Cellular and Infection Microbiology* **10** 470 (2020).

3-3) - Can the authors describe the impact of the higher refractive index and optical losses of the silicon layer on the top PSRP sensor and its sensitivity compared to more conventional dielectric substrates for the metal film?

In Fig L2 below we plot simulation data to demonstrate the impact of the higher refractive index. Three different substrates are shown; high refractive index glass ($n = 1.7746$, $k = 0$), amorphous silicon ($n = 4.089$, $k = 0.087$), and a hypothetical “lossless amorphous silicon” ($n = 4.089$, $k = 0$). In all cases the wavelength is 670 nm, the dielectric thickness is 110 nm, and the device is coated with water ($n = 1.333$). Figures L2a and L2b have gold thickness of 30 nm. Figures L2c and L2d have gold thickness of 50 nm.

Evidently, the optical losses of silicon greatly affect the reflectance due to absorption of incoming and reflected light. [Note that this absorption is compounded by the multiple reflection of cavity modes.] As such, while our device could still be measured in a conventional angle sweep of reflection, its performance would be noticeably worse than a conventional dielectric. However, the higher refractive index and optical losses have a much more limited effect on the PSRP in the gold film as can be seen in the plot of $|E|^2$ at the top of the gold layer (Figs. L2b and L2d). Little qualitative difference is seen between the angular dependence of the electric field for all three types of substrate with the silicon substrate having a magnitude of 74% that of the glass substrate for gold thickness of 30 nm and 94% for gold thickness of 50 nm.

We therefore conclude that the higher refractive index and optical loss of silicon does not significantly impact the plasmon within the gold layer.

Fig. L2: Reflectance and $|E|^2$ at the top of the gold layer for gold thickness 30 nm (a and b) and 50 nm (c and d).

3-4) - Please elaborate on available evidence and the meaning of the following sentence: "with the hot electron-based device having a smaller working range of RI than the direct photogeneration based device with the latter varying monotonically over the entire range of RI". From Figure 3c, I just see a lower photocurrent for the TiO2 device, but the monotonically decreasing current occurs for both devices. The exact dependence on concentration depends on the angle chosen, but the shapes of figures 3a and 3b are qualitatively similar to a first approximation.

We thank the reviewer for their comment and agree that our sentence and figure were not sufficiently clear. We have made two changes to this figure; the first is to multiply the data for TiO₂ by a factor of 10, the second to replace the Si data with data for gold thickness of 30 nm. These two changes now more clearly illustrate the different behaviour of the two devices. On further consideration we agree with the reviewer that it is not particularly insightful or noteworthy to discuss the *monotonical* nature over the arbitrary range of RI (1.333 to 1.397) because a smaller range of RI should show monotonical change for both types of device. We have therefore removed the word *monotonical* and instead describe only the larger working range of the silicon device (64 mRIU) compared to the TiO₂ device (38 mRIU).

3-5) - The work is succinct and simple to understand but unnecessarily difficult to read, in part because of lack of information in figures and their captions. For example, in Figure 2, the meaning of the ON/OFF ratio could be introduced in more detail for clarity. Also in Figure 2, the wavelengths and thickness of the dashed line layer are not described. Another example: a legend in Figure 3c could save time to the reader by indicating which device it corresponds to.

We apologise for the difficulty in reading the previous manuscript and have made several changes to improve this. Regarding the ON/OFF ratio, as well describing this more carefully in the text, we also change the presentation of the data in Fig. 1c to help make this definition more intuitive. As requested, we have added legends and other details to the figures and figure captions to save the time of the reader.

3-6) - Indicate the meaning of the dashed line in Figure 1c, as the origin of the x-axis and the thickness of the metal film are not described in the caption.

We apologise for this oversight and have added these details to the figure captions.

REVIEWER COMMENTS

Reviewer #1 (Remarks to the Author):

Dear Authors,
I have read the revised paper and have no more comments.

Regards

Reviewer #2 (Remarks to the Author):

Thanks for addressing my comments. The manuscript is much better in terms of its presentation. However, technical flaws remain. My comments on the need for intensity calibration and the lack of peak fitting benefit are not adequately addressed. Fig. 4 in the revised manuscript does show fixed angle measurements. However, how is the signal referenced? Are the measurements repeatable? Enhanced sensitivity claimed here does not seem to correlate with the Fabry-Perot+SPR approach. A control experiment with just SPR or Fabry-Perot sensing would be more convincing.

Reviewer #3 (Remarks to the Author):

The authors have addressed my previous concerns.

As a final comment, the optoelectronic implementation of a plasmonic biosensor is potentially interesting but faces one important hurdle in my opinion: the bottleneck for applications is not integrated readout (with a bulky optical device) but chemical functionalization requiring a chemistry laboratory to prepare each sensor. Substrates cannot be reused for sensing, so single-use optoelectronic devices would be required. The setup still needs an electronic device for readout, so part of the potential benefit of an optoelectronic sensor disappears.

Response to reviewers

On behalf of all authors I wish to express our deep gratitude to the reviewers for their time and effort in re-evaluating our manuscript. We are pleased that the reviewers approved of the changes made to the manuscript and we once again thank them for their initial comments that helped immensely in improving our manuscript. In this document we wish to address the most recent comments made by the reviewers. Remarks from reviewers are copied in italic font and our responses are given in roman font. At the end of this document we list all changes that have been to the manuscript.

Reviewer #1 (Remarks to the Author):

*Dear Authors,
I have read the revised paper and have no more comments.*

Regards

We thank the reviewer once more for their highly useful comments.

Reviewer #2 (Remarks to the Author):

Thanks for addressing my comments. The manuscript is much better in terms of its presentation. However, technical flaws remain. My comments on the need for intensity calibration and the lack of peak fitting benefit are not adequately addressed. Fig. 4 in the revised manuscript does show fixed angle measurements. However, how is the signal referenced? Are the measurements repeatable? Enhanced sensitivity claimed here does not seem to correlate with the Fabry-Perot+SPR approach. A control experiment with just SPR or Fabry-Perot sensing would be more convincing.

We thank the reviewer once more for their valuable comments. We first address the question of the surface plasmon resonance (SPR) and Fabry-Pérot (FP) approach and then comment on the referencing and repeatability of our sensor.

As requested by the reviewer we have performed control experiments to illustrate the FP and SPR effects and compare them with simulations (see Fig. L1 & L2). For the purpose of switching on and off the FP, and in consideration of Fig. 2 of the manuscript, we changed the thickness of the silicon layer (110 nm and 150 nm). For the purpose of switching on and off the SPR effect we use the gold thickness of 30 nm used in the paper and a gold thickness of 150 nm. In the latter case the gold thickness is too thick to support surface plasmon resonance. Alternatively another non-plasmonic metallic material could have been used, but we decided to use a thicker gold layer for this comparison in order to keep all four types of devices as similar as possible. (A different metallic material would likely have both different electronic, due to different work functions, and different surface modification properties for ligand immobilization such as antibodies, due to different surface chemistry.)

Each figure is a plot, similar to Figure 3 in the manuscript, of the angular dependence of the photocurrent with different concentrations of glycerol. Figure a) shows the data when the device supports both SPR and FP modes. Figure b) supports the SPR, but not the FP mode.

Figure c) supports the FP mode, but not the SPR mode. Figure d) supports neither mode. Clearly only devices supporting SPR modes are sensitive to the RI of the solution and furthermore coupling to the FP mode clearly enhances the current sensitivity to changes of RI. The current is larger in figure c than figure a (likewise for figures d and b) because the thicker metal film reflects more light back into the silicon layer. The experimental measurements are in good agreement with simulations.

Fig. L1: Experimental Refractive Index sensing with and without SPR and FP effects.

Photocurrent as a function of angle in solutions of different glycerol concentration [0%, 10%, 20%, 30%, 40%, 50%]. In a) and b) SPR is supported by a 30 nm gold film. In c) and d) SPR is not supported by a 150 nm gold film. In a) and c) the FP effect is supported by a 110 nm silicon layer. In b) and d) the FP effect is not supported by a 150 nm silicon layer.

Fig. L2: Simulated Refractive Index sensing with and without SPR and FP effects. CA as a function of angle in solutions of different glycerol concentration [0%, 10%, 20%, 30%, 40%, 50%]. In a) and b) SPR is supported by a 30 nm gold film. In c) and d) SPR is not supported by a 150 nm gold film. In a) and c) the FP effect is supported by a 110 nm silicon layer. In b) and d) the FP effect is not supported by a 150 nm silicon layer.

Regarding the understandable questions of referencing and repeatability we would first like to outline how we propose the device would be used in practice. The single gold electrode presented in this study will be replaced with two separate and identical electrodes. Electrode **A** is coated with antibodies in the same manner as in the manuscript and electrode **B** is covered with dummy antibodies (or left uncoated). Using either a wide beam or split beam laser the two electrodes will be measured concurrently and both electrodes will be exposed to the same solution at the same time. Thus only the current through electrode **A** should change due to antigen-antibody interactions and the current through electrode **B** will be used as a reference for unintended variables such as fluctuations in laser power.

Finally we turn once more to the suitability of fixed angle operation. To this end we refer to a new section that we have added to the supplementary information (Supplementary Note 6: Distinguishing between changes of signal due to bulk and interaction effects). Ideally there should be no changes to the bulk RI when exchanging the buffer solution with the antigen containing solution, however, in practicality small changes are difficult to eliminate entirely. In the supplementary information we show that fitting the current minimum on its own is insufficient to determine whether the change of refractive index, ΔRI , is caused by true antibody-antigen interactions in the vicinity of the gold layer or simple bulk changes of the solution. To distinguish between interaction and bulk effects would require determining not just the shift of the current minimum, but also the shift, if any, of the total internal reflection angle. This is by no means unfeasible, however, the fixed angle operation gives a straightforward, compact alternative. The fixed angle operation requires a single fit of the logarithmic time dependence that takes advantage of the antibody-antigen interaction time. In our experiments (Fig. 4 and Supplementary Fig. 5) we confirmed that the timescale of this interaction is the same for both our sensor and the BLItz, thus we are confident that this method is suitable for detecting antibody-antigen interactions. In the "Sensing of antigen-antibody interaction" section of the manuscript we have inserted a new sentence and edited another to introduce this new supplementary information.

Regarding the repeatability of our device, each error bar in Figs. 2 and 4 is calculated using the standard deviation of four devices. We consider that the main sources of variation in the change of current, δI , in Fig. 4 is a matter of the basic sample fabrication techniques employed. The prototype used in the manuscript is based on a simple intrinsic silicon/gold Schottky junction. It is well known that the Schottky junction is susceptible to charge states that can cause variation from sample to sample. There is, however, a wealth of literature from the solar cell and other communities regarding silicon-gold interfaces that we are confident will improve the consistency of our device. This remains a work in progress and one promising approach involves replacing the purely *i*-type silicon layer with an *n-i* junction to stabilise the electrical properties. In addition to these design and fabrication issues, the constant angle measurements of Fig. 4 were performed on a rotating stage so there may be a small misalignment in the angles used in each experiment creating another source of variation in δI . This can be easily eliminated by switching to a custom made fixed angle stage.

Reviewer #3 (Remarks to the Author):

The authors have addressed my previous concerns.

As a final comment, the optoelectronic implementation of a plasmonic biosensor is potentially interesting but faces one important hurdle in my opinion: the bottleneck for applications is not integrated readout (with a bulky optical device) but chemical functionalization requiring a chemistry laboratory to prepare each sensor. Substrates cannot be reused for sensing, so single-use optoelectronic devices would be required. The setup still needs an electronic device for readout, so part of the potential benefit of an optoelectronic sensor disappears.

We thank the reviewer once more. Your comments have been very useful for our next research goal. Thank you.

List of changes to the manuscript
(Changes made to the main text and figure captions are highlighted in red text)

Affiliation #4 changed due to renaming of university:

Center for Emergent Functional Matter Science, National Yang Ming Chiao Tung University, Hsinchu 30010, Taiwan

Corresponding author email address changed due to company reorganisation:

giles.allison@imra-japan.com

Initialisations have been removed from Abstract.

Name of virus corrected in Introduction:

The recent pandemic caused by the novel coronavirus (SARS-CoV-2) has

Sentence added to “Sensing of antigen-antibody interaction” section of manuscript:

Time dependence measurements in fixed angle mode of operation allow for simple confirmation that the change of refractive index detected by the sensor is due to antigen-antibody interactions and not simple fluctuations in the bulk RI of the different solutions used in the experiment (see supplementary information).

Sentence edited in “Sensing of antigen-antibody interaction” section of manuscript:

By continuously monitoring the current in the fixed angle mode of operation, not only can we confirm that the change is due to antigen-antibody interaction, but we can also determine the affinity, concentration, and binding kinetics of this interaction.

Figure 3 caption: typo corrected in list of RI:

... with RI = 1.333, 1.346, 1.359, 1.371, 1.384, 1.397 ...

Reference #10 updated from pre-print to published article. (Typos have been corrected in other references.)

Some additional details have been added to the methods section.

Data availability statement added:

Source data are provided with this paper.

Funding sources added to Acknowledgements:

... HM acknowledges the financial support from JSPS KAKENHI (Grants JP18H05205). This work was supported by IMRA Japan Co., Ltd.

Conflict of interests section renamed to Competing Interests and information regarding patents has been updated.

All figures have been reformatted to comply with submission guidelines (font sizes, colours etc.). The content of all figures remains the same as in the previous version.

Text (including supplementary information) has been reformatted to submission guidelines.

Supplementary Note 6: Distinguishing between changes of signal due to bulk and interaction effects added to supplementary information.

Supplementary notes have been numbered and re-ordered to correspond to the order stated in the main text.

REVIEWERS' COMMENTS

Reviewer #2 (Remarks to the Author):

Thanks for revising the manuscript. The control experiments do show the advantage of the technique clearly and have convinced that the presented work is of technological importance.

Response to reviewers

On behalf of all authors I wish to express our deep gratitude to all reviewers for their time and effort spent evaluating our manuscript. We truly believe that their comments have been of great benefit in enabling us to improve our manuscript. In this document we address the most recent comments made by reviewer #2. The comments of reviewer #2 are copied in italic font and our response is given in roman font.

Reviewer #2 (Remarks to the Author):

Thanks for revising the manuscript. The control experiments do show the advantage of the technique clearly and have convinced that the presented work is of technological importance.

We thank the reviewer once more for their valuable comments throughout the review process. We are pleased that the reviewer agrees with the advantage of the technique and finds our work of technological importance.